# Outcomes and Clinical Characteristics of COVID-19 in Patients with Tuberculosis: A Retrospective Matched Cohort Study

Zachary Hartnady [1], Benjamin Krehbiel [1], Ashley Stenzel [2] and David Tierney [1,*]

1   Department of Medical Education, Abbott Northwestern Hospital, Minneapolis, MN 55407, USA;
    zachary.hartnady@allina.com (Z.H.)
2   Care Delivery Research, Allina Health, Minneapolis, MN 55407, USA
*   Correspondence: david.tierney@allina.com

**Abstract:** The outcomes and characteristics of acute coronavirus disease 2019 (COVID-19) infection in patients with tuberculosis (TB) represent an evolving area of literature. This retrospective cohort study (March 2020–January 2021) within a large United States health system evaluated clinical and demographic characteristics, illness severity, complications, and mortality associated with acute COVID-19 infection in patients with TB (n = 31) compared to a matched (1:3) COVID-19 cohort without TB (n = 93). In the COVID-19 + TB cohort, TB was active in 32% and latent in 65% of patients, most patients (55%) had pulmonary TB, and 68% had previously undergone treatment for their TB. Patients with COVID-19 + TB infection had higher rates of hospitalization (45% vs. 36%, $p = 0.34$), intensive care unit (ICU) stay (16% vs. 8%, $p = 0.16$), and need for mechanical ventilation (13% vs. 3% $p = 0.06$). Discordant with those higher rates of markers typically denoting more severe illness, TB patients with acute COVID-19 did not have longer length-of-stay (5.0 vs. 6.1 days, $p = 0.97$), in-hospital mortality (3.2% vs. 3.2%, $p = 1.00$), or 30-day mortality (6.5% vs. 4.3%, $p = 0.63$). This study, while having limitations for extrapolation, cautions the notion that patients with COVID-19 and TB infers worse outcomes and adds to the growing body of literature on the interaction between these two infections.

**Keywords:** COVID-19; coronavirus; tuberculosis; cohort study; clinical outcomes

## 1. Introduction

The coronavirus disease 2019 (COVID-19) pandemic changed the landscape of medical practice and challenged many health systems in the United States and abroad. Surging numbers of respiratory illness patients, stress on staffing, and allocation of critical resources were all challenges of the early COVID-19 pandemic. Case fatality rates in the early pandemic were estimated to be around 4.0% globally [1], with significant variation between countries, ranging from 0 to 20% [1]. A large component of variation in mortality risk can be attributed to increasing age as well as other comorbid medical conditions such as diabetes, cardiovascular disease, and chronic obstructive pulmonary disease (COPD) [2]. Identifying risk factors for severe COVID-19 infection can allow for informed discussion around prognosis between the patient and medical provider. These discussions, especially around ICU-level care and need for mechanical ventilation, can better inform patients and their families prior to the worsening of COVID-19 disease.

One comorbidity that has had an increasing amount of research is a co-respiratory infectious disease in tuberculosis (TB). The intersection of these pandemics has provided many challenges in resource allocation for tuberculosis control, public health focus, pharmaceutical research and development, and other epidemiological metrics in controlling the global spread of TB, the former leading cause of death from a single infectious agent [3,4]. Many patients with primary pulmonary TB can develop residual lung pathology that can lead to obstructive lung disease, bronchiectasis, and increased risk for non-tuberculous infections [5]. It has been postulated that immune interactions between COVID-19 and TB can

increase the risk of TB reactivation [6,7] as well as alter disease severity of COVID-19 [8,9]. Early case series would suggest the presence of TB increases the risk of developing severe COVID-19 infection [10,11], prompting further evaluation by large health departments internationally. The Centers for Disease Control and Prevention (CDC) published a brief review analyzing five cohort studies and one ecologic study, all in 2020 prior to the COVID-19 vaccine data, to examine the interactions of TB and COVID-19 [12]. The conclusion of the CDC review was that tuberculosis was associated with increased COVID-19 mortality based on hazard, risk, and odds ratios of the four studies analyzed [12]. However, this review failed to analyze other patient-centered metrics such as need for ICU-level care, mechanical ventilation, readmission rates, and length of stay.

The aim of this study was to compare outcomes including level of care, mechanical ventilation, length of stay, or mortality for matched cohorts of COVID-19 with and without TB.

## 2. Materials and Methods

### 2.1. Study Design and Patient Population

This was a retrospective cohort study, including patients seen within a 12-hospital, 98-clinic health system in Minnesota, United States. Patients with a positive COVID-19 polymerase chain reaction (PCR) test between 1 March 2020 and 14 January 2021 were included in the dataset from both the inpatient and outpatient settings. The unmatched, initial dataset included all patients with a history of TB (new or prior diagnosis), as well as those with no documented history seen in our health system between the dates. The initial dataset included a total of 8744 patients. Patients with TB were then matched (3:1) to patients without TB by age and sex, with the final dataset including 93 patients without TB, and 31 patients with a TB diagnosis.

### 2.2. Measures and Data Collection

All data were abstracted from electronic health records. The independent variable of interest was TB status (positive, negative) at COVID-19 encounter. The outcomes of interest included level of care required for COVID-19 (outpatient, inpatient), any intensive care unit (ICU) admission, mechanical ventilation, multiple hospitalizations, in-hospital death, total hospital length of stay (LOS), and vital status at 30 days post-discharge (alive, deceased). Additionally, an outcome of interest included COVID-19 complications, which consisted of deep venous thrombosis (DVT), pulmonary embolism (PE), non-COVID-19 infection, sepsis, and stroke. Other measures included in this study were age, sex, race, ethnicity, smoking history, cardiopulmonary comorbidities (asthma, COPD, obstructive sleep apnea, interstitial lung disease, congestive heart failure, and coronary artery disease). Among those with a history of TB, data were collected regarding TB status (latent vs. active), TB type (pulmonary, extrapulmonary, or both), and history of TB treatment.

### 2.3. Statistical Analysis

Descriptive statistics were used to examine the distribution of clinical and demographic characteristics among the total study population. Chi square, Fisher's exact, Wilcoxon rank sum, and Student's *t*-tests were used, as appropriate, to compare characteristics among study groups. Likewise, the aforementioned statistical tests were used to examine differences in patient outcomes by TB status. All analyses were carried out using SAS software version 9.4 (Cary, NC); *p*-values < 0.05 were considered statistically significant.

This study was determined to be exempt by the Allina Health Institutional Review Board (IRB #1661950).

## 3. Results

A total of 124 patients were included in the study as displayed in Table 1: 93 patients with COVID-19 and 31 patients with TB and COVID-19 matched 3:1 in non-TB and TB

age/sex matched cohorts, respectively. Mean age was 57 years and 65% of the patients were female in the matched cohorts. A higher proportion of the TB cohort identified as Black/African American race (52% vs. 14%), and the majority of the non-TB cohort identified as white (73% vs. 32%). There was a higher proportion of underlying cardiopulmonary comorbidities in the TB cohort (77% vs. 43%, *p* = 0.01). When cardiac vs. pulmonary comorbid conditions were analyzed individually, there was a statistically significant level of pulmonary conditions in the TB group driving the composite characteristic (71% vs. 21%, *p* < 0.001).

**Table 1.** Distribution of demographic and clinical characteristics among patients with coronavirus disease 2019 (COVID-19), by tuberculosis (TB) status.

| Characteristic | Total (N = 124) Mean (SD) | TB Negative [a] (n = 93) Mean (SD) | TB Positive (n = 31) Mean (SD) | *p*-Value |
|---|---|---|---|---|
| Age, years | 57.3 (17.5) | 57.3 (17.6) | 57.3 (18.8) | 1.00 |
| | **n (%)** | **n (%)** | **n (%)** | |
| Sex | | | | |
| Female | 80 (64.5%) | 60 (64.5%) | 20 (64.5%) | 1.00 |
| Male | 44 (35.5%) | 33 (35.5%) | 11 (35.5%) | |
| Race | | | | |
| American Indian/Alaskan Native | 2 (1.6%) | 1 (1.1%) | 1 (3.2%) | |
| Asian | 8 (6.5%) | 4 (4.3%) | 4 (12.9%) | |
| Black/African American | 29 (23.4%) | 13 (14.0%) | 16 (51.6%) | <0.0001 |
| Native Hawaiian or other Pacific | 1 (0.8%) | 1 (1.1%) | 0 (0.0%) | |
| White | 78 (62.9%) | 68 (73.1%) | 10 (32.3%) | |
| Unknown | 6 (4.8%) | 6 (6.5%) | 0 (0.0%) | |
| Ethnicity | | | | |
| Hispanic | 15 (12.1%) | 11 (11.8%) | 4 (12.9%) | |
| Non-Hispanic | 107 (86.3%) | 80 (86.0%) | 27 (87.1%) | 1.00 |
| Unknown | 2 (1.6%) | 2 (2.2%) | 0 (0.0%) | |
| Smoking history | | | | |
| Never/Unknown | 74 (59.7%) | 56 (60.2%) | 18 (58.1%) | 0.83 |
| Any smoking history | 50 (40.3%) | 37 (39.8%) | 13 (41.9%) | |
| Cardiopulmonary comorbidity [b] | | | | |
| None known | 60 (48.4%) | 53 (57.0%) | 7 (22.6%) | 0.001 |
| Any | 64 (51.6%) | 40 (43.0%) | 24 (77.4%) | |
| Cardiac comorbidities [c] | | | | |
| None known | 100 (80.7%) | 76 (81.7%) | 24 (77.4%) | 0.60 |
| Any | 24 (19.4%) | 17 (18.3%) | 7 (22.6%) | |
| Pulmonary comorbidities [d] | | | | |
| None known | 70 (56.5%) | 61 (65.6%) | 9 (29.0%) | 0.0004 |
| Any | 54 (43.6%) | 32 (34.4%) | 22 (71.0%) | |
| TB status | | | | |
| Latent | NA | NA | 20 (64.5%) | NA |
| Active | | | 10 (32.3%) | |
| Unknown | | | 1 (3.2%) | |
| TB type | | | | |
| Pulmonary | | | 17 (54.8%) | |
| Extrapulmonary | NA | NA | 7 (22.6%) | NA |
| Both | | | 4 (12.9%) | |
| Unknown | | | 3 (9.7%) | |
| History of pulmonary TB treatment | | | | |
| No/unknown | NA | NA | 10 (32.3%) | NA |
| Yes | | | 21 (67.7%) | |

[a] Matched to TB-positive patients on age and sex. [b] Includes: asthma, chronic obstructive pulmonary disease, obstructive sleep apnea, interstitial lung disease, congestive heart failure, and coronary artery disease. [c] Includes: congestive heart failure and coronary artery disease. [d] Includes: asthma, chronic obstructive pulmonary disease, obstructive sleep apnea, and interstitial lung disease.

For the TB patients, 32% of patients had an active infection vs. 65% with latent infections. The majority of patients in the TB cohort had pulmonary TB (55%) compared to extra-pulmonary (23%) TB, and 13% of patients had evidence of pulmonary and extra-pulmonary TB. TB distribution was unknown in 10% of patients. Of the patients who had pulmonary TB, 68% had a history of past treatment for TB compared to 32% without past treatment. The 10 patients with active TB were further subdivided into the following groups: simultaneous new diagnosis of TB with COVID-19 (n = 1; TB meningitis), reactivation of latent TB (n = 6), and on treatment for active TB when acquired COVID-19 (n = 3).

Table 2 displays various outcomes of interest. There was no significant difference in highest level of care between cohorts as represented by a *p*-value of 0.34 utilizing a Chi-squared test. Specifically, 65% of patients with COVID-19 and 55% of patients with TB + COVID-19 were treated as outpatients, contrasted to 36% of patients with COVID-19 vs. 45% of patients with TB + COVID-19 treated as inpatients. There was a trend towards a higher rate of ICU care (16% vs. 7.5%, *p* = 0.16) and need for mechanical ventilation (13% vs. 3%, *p* = 0.06) in the 31 patients with TB when compared to the 93 with COVID-19 alone. There was no significant difference for in-hospital (3.2% vs. 3.2%, *p* = 1.00) or 30-day mortality (6.5% vs. 4.3%, *p* = 0.63) for the TB and non-TB cohorts, respectively. Hospital LOS did not differ significantly between groups (6.1 days vs. 5.0 days, *p* = 0.97).

**Table 2.** Distribution of COVID-19 care-related characteristics and outcomes among patients with COVID-19, by tuberculosis (TB) status.

| Characteristic | Total (N = 124) n (%) | TB Negative [a] (n = 93) n (%) | TB Positive (n = 31) n (%) | *p*-Value |
|---|---|---|---|---|
| Highest level of care for COVID-19 | | | | |
| Outpatient | 77 (62.1%) | 60 (64.5%) | 17 (54.8%) | 0.34 |
| Inpatient | 47 (37.9%) | 33 (35.5%) | 14 (45.2%) | |
| Required ICU care | | | | |
| No | 112 (90.3%) | 86 (92.5%) | 26 (83.9%) | 0.16 |
| Yes | 12 (9.7%) | 7 (7.5%) | 5 (16.1%) | |
| Required mechanical ventilation | | | | |
| No | 117 (94.4%) | 90 (96.8%) | 27 (87.1%) | 0.06 |
| Yes | 7 (5.7%) | 3 (3.2%) | 4 (12.9%) | |
| Required multiple hospitalizations | | | | |
| No | 115 (92.7%) | 86 (92.5%) | 29 (93.6%) | 1.00 |
| Yes | 9 (7.3%) | 7 (7.5%) | 2 (6.5%) | |
| Complications [b] | | | | |
| Deep venous thrombosis (DVT) | 1 (0.8%) | 1 (1.1%) | 0 (0.0%) | 1.00 |
| Pulmonary embolism (PE) | 2 (1.6%) | 2 (2.2%) | 0 (0.0%) | 1.00 |
| Non-COVID-19 infection | 7 (5.7%) | 4 (4.3%) | 3 (9.7%) | 0.37 |
| Sepsis | 3 (2.4%) | 1 (1.1%) | 2 (6.5%) | 0.15 |
| Stroke | 2 (1.6%) | 1 (1.1%) | 1 (3.2%) | 0.44 |
| In-hospital death | | | | |
| No | 120 (96.8%) | 90 (96.8%) | 30 (96.8%) | 1.00 |
| Yes | 4 (3.2%) | 3 (3.2%) | 1 (3.2%) | |
| Vital status at 30 days post-discharge | | | | |
| Alive | 118 (95.2%) | 89 (95.7%) | 29 (93.6%) | 0.63 |
| Deceased | 6 (4.8%) | 4 (4.3%) | 2 (6.5%) | |
| | Median (range) | Median (range) | Median (range) | |
| Total hospital length of stay, days [c] | 5.3 (1.5–46.3) | 6.1 (1.5–30.0) | 5.0 (1.9–46.3) | 0.97 |

[a] Matched to TB-positive patients on age and sex. [b] Within 30 days of admission for COVID-19. [c] Sum of days for all COVID-19 hospitalizations per patient. ICU = intensive care unit; TB = tuberculosis.

## 4. Discussion

In this retrospective matched cohort study, a (+) COVID-19 PCR and history of TB was associated with higher rates of inpatient and ICU-level care as well as mechanical ventilation (13% compared to 3%, *p* 0.06); however, despite the increase in these severity

markers, a history of TB did not correlate with a higher in-hospital or 30-day mortality rate, nor with increased LOS. Surprisingly, the presence of TB as a comorbidity in patients with COVID-19 disease did not worsen these clinical metrics. This would suggest that TB may not be as impactful on COVID-19 disease severity as previously hypothesized. This finding is even more surprising given the fact that the TB cohort also had a significantly higher rate of cardiopulmonary comorbidities felt to be negatively impactful on COVID-19 outcomes compared to the non-TB group.

Various mechanisms have been proposed for how TB could alter COVID-19 disease severity, with a focus on the altered Th1 response in TB creating an unfavorable immunologic microenvironment for COVID-19 leading to exaggerated immune responses [13]. This study's lack of difference in outcomes among patients with concomitant TB is contrary to the limited previous literature on this topic, specifically from the CDC in 2020, although there is likely a significant difference in patient populations in our study versus those in the CDC review [12]. The importance of this finding relates to the uncertain clinical influence the TB comorbidity should have on health care providers' prognostication and discussions with TB patients and their families in the setting of COVID-19 infection. From our small cohort study within a very large United States health system, clinicians should not use the presence of TB as a negative predictor for COVID-19 outcomes. Further, this information is important in limited resource settings where COVID-19 treatment allocation may be stratified based on predictors of survival.

Limitations of this study include the small cohort size but it is strengthened by the 3:1 matched study design. Similarly, TB seen in Minnesota patient populations may not be representative of global TB/COVID-19 interactions. In addition, most patients in the pulmonary TB group had a history of TB treatment, potentially minimizing the effect of previous infection on COVID-19 severity given typical treatment effectiveness [14]. In addition, the number of patients who were receiving TB treatment at the time of COVID-19 diagnosis was unknown, possibly leading to suppression of TB disease and decreasing TB–COVID-19 interactions. Ideally, the association between COVID-19 outcomes and individual TB patient phenotypes (e.g., active pulmonary vs. latent pulmonary) would be analyzed; however, the limited size of this cohort population makes that impracticable and an area for future research. This study is unable to account for additional confounding beyond the variables included in our matching.

Our results add an alternate conclusion to the existing data on outcomes in patients with TB and COVID-19 and may caution the use of TB as a negative predictor for outcomes in COVID-19 disease until further research is performed. These data can be used to discuss prognosis and inform patient-centered goals-of-care discussions when confronted with this unique circumstance of co-infection. More research is needed on how TB and COVID-19 interact within this subgroup of patients in the United States and abroad presenting with both diseases that intuitively should have worsened outcomes. Specific research into immunologic interactions between the two respiratory illnesses and broader population studies can provide insight into how COVID-19 and TB impact patients worldwide.

**Author Contributions:** Conceptualization, D.T.; methodology, D.T. and A.S.; software, D.T. and A.S.; validation, D.T., Z.H., B.K. and A.S.; formal analysis, B.K. and A.S.; investigation, D.T.; resources, D.T. and A.S.; data curation, A.S.; writing—original draft preparation, Z.H. and B.K.; writing—review and editing, D.T., Z.H., B.K. and A.S.; visualization, D.T.; supervision, D.T.; project administration, D.T.; funding acquisition, D.T. All authors have read and agreed to the published version of the manuscript.

**Funding:** This research received no external funding.

**Institutional Review Board Statement:** The study was conducted in accordance with the Declaration of Helsinki, and approved by the Institutional Review Board (or Ethics Committee) of Allina Health protocol code 1661950 as an exempt study on 30 November 2020.

**Informed Consent Statement:** Patient consent was waived due to exempt category 4—secondary research for which consent is not required: information is recorded by the investigator in such a manner that the identity of subjects cannot readily be ascertained directly or through identifiers linked to the subjects and the investigator does not contact or re-identify the subjects.

**Data Availability Statement:** The data presented in this study are available on request from the corresponding author, depending on restrictions, e.g., privacy or ethical. The data are not publicly available due to private health information restrictions within the Allina Health system.

**Acknowledgments:** All authors have read and agreed to the published version of the manuscript.

**Conflicts of Interest:** The authors declare no conflict of interest.

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
