# Peer review of "Outcomes and Clinical Characteristics of COVID-19 in Patients with Tuberculosis: A Retrospective Matched Cohort Study"

_2036-7449, doi:10.3390/idr15020021_

Round 1

Reviewer 1 Report

1- I suggest separating comorbidities  in pulmonary and cardiovascular, not leaving as cardiopulmonary, because the pathologies overlap and leads to the error in tabla 1 where describes pulmonary and cardiac pathologies and c and d are not defined at the base of the table.

2- Table 2 does not indicate the site of index d

3- Most patients had latent and previously treated TB, I recommend reviewing how many of the patients had reactivation of TB disease. It should be used to give this data, since it is know That COVID19 favors the progression of TB infection. Reinforcing the concept is a good contribution.

4- I believe that a general analysis of patients with TB of diferent clinical presentation and with different inflamatory response, as in this case (active, pulmonary, extrapulmonary, latente and previously treated)against another pathology, requires to analyze each one separately.

5- It is not said how many patients had active pulmonary TB with positive smear at study entry and comparison with patients with COVID19, a grup that may presentwith association as a risk factor for severe disease due to COVID19

6- In the discussion, the comparison with CDC results is not valid, since they are different group of patients. The comparison can given, but making the clarification  

Author Response

Please see the attachment - Responses to Reviewer 1 comments found in blue text and itemized 1-6

Reviewer 2 Report

General comments

The manuscript provides an interesting and timely study on the outcomes of COVID-19 in patients with comorbid tuberculosis. With some revisions as suggested above, this paper could be an important contribution to the literature.

Also, there are several long sentences in the manuscript that can be improved for clarity:

  • Sentence 1: Early case fatality rates were estimated around 4.0% globally with tremendous variation between countries, ranging from 0 to 20%.
  • Sentence 4: Identifying these risk factors for severe COVID-19 can allow for informed prognostic and goals-of-care discussions between the patient and medical provider, especially as they pertain to ICU-level care, mechanical ventilation, and other clinical questions with implications for individual prognosis.
  • Sentence 12: Patients who were seen either inpatient or outpatient with a positive COVID-19 polymerase chain reaction (PCR) test between March 1, 2020 and January 14, 2021 were included in the dataset.

The abbreviation "COPD" is defined at first use, but "ICU" and "LOS" are not. "DVT" and "PE" are abbreviations that are not defined at first use.

Specific comments

Abstract

The abstract provides a clear overview of the study, its objectives, methodology, and main findings. The language is concise and easy to understand. However, there are a few grammatical and structural issues that need to be addressed:

  • Line 9: Change "undescribed" to "understudied" to better reflect the context of the research.
  • Line 10: Add a hyphen between "illness" and "severity" to make it a compound adjective.
  • Line 12: Replace "in- and outpatients" with "inpatients and outpatients" to improve clarity.
  • Line 14: Add "with" before "new or previous diagnosis" to improve sentence structure.
  • Line 19: Remove the period after "mor" and capitalize the "M" in "mortality."
  • Line 22: Change "this did not correlate with higher mortality nor increased LOS" to "these factors were not associated with higher mortality or increased length of stay."

Introduction: The introduction provides a good overview of the topic and highlights the research gap in the relationship between COVID-19 and tuberculosis. However, it could benefit from some minor revisions. For instance, please add the citation to reference [1] to support the claim of the global case fatality rate estimation of 4.0% except it has the same source as the 0-20% range. Additionally, some sentence constructions can be improved for clarity.

  • Line 28: It would be helpful to provide a brief summary of how COVID-19 has impacted the healthcare system before moving on to case fatality rates.
  • Lines 40-45: The discussion on TB and its prevalence can be a bit lengthy and could be condensed to focus on the specific topic of the study.
  • Line 47: Instead of just mentioning the CDC review, it would be helpful to briefly summarize the main findings and limitations of the review.

Methods: The methods section is comprehensive and clearly described. However, some information can be added to enhance the paper. Specifically, information about how matching was performed to select control patients can be included. Also, the definition and diagnostic criteria of the cardiopulmonary comorbidities, as well as the COVID-19 complications, should be stated. Also the STROBE guideline for observational study has not been adhered to and the authors should download this and follow the STROBE Guideline

  • Line 56: The study design should be introduced earlier in the section.
  • Lines 60-61: It would be helpful to specify how the matching process was done (e.g., matching algorithm, specific criteria used).
  • Lines 64-75: The measures and data collection section is quite detailed, but it is missing information on the specific statistical methods used. It would be helpful to briefly describe the statistical tests and models used in the analysis.

Results: The results section is well organized and provides a good summary of the findings. However, some improvements can be made. For instance, please add the total number of patients included in the study and the proportion of patients with TB. Also, the authors should indicate whether the patients with TB were receiving any TB treatment at the time of the COVID-19 diagnosis.

  • Lines 92-95: The sentence is a bit confusing, it may be helpful to rephrase it.
  • Line 100: The p-value of 0.043 should be reported with its corresponding statistical test.
  • Lines 101-102: The sentence is a bit difficult to follow. It may be helpful to break it up into two sentences and provide more context on what the p-value represents.
  • Lines 105-106: It would be helpful to provide the exact number of patients in each group (with and without TB).
  • Lines 114-119: The discussion on limitations is brief and could be expanded upon to better contextualize the findings.

Discussion: The discussion is well written and provides a comprehensive interpretation of the study results. The authors could add some additional information about the possible biological mechanisms that may explain the association between TB and severe COVID-19 disease.

  • Line 135: The discussion on the findings is brief and could be expanded upon to better contextualize the results in relation to previous literature.
  • Lines 139-142: It may be helpful to provide a more detailed explanation of the potential mechanisms behind the association between TB and severe COVID-19 disease.
  • Lines 144-147: The implications of the study could be discussed in more detail, including the potential impact on clinical practice and future research directions.

Author Response

Please see the attachment - responses to Reviewer 2 identified in blue text itemized by Section as described in reviewer response. STROBE guidelines included at end of document in a table format.

Reviewer 3 Report

In the present manuscript "Outcomes and Clinical Characteristics of COVID-19 in Patients with Tuberculosis: A Retrospective Matched Cohort Study" the authors have discussed the COVID-19 outcomes in the TB patient. The report briefly describes that history with TB did not relate to the COVID-19-related in-hospital or 30-day mortality rate or the length of stay. 

However, the evidence described in the present study is not enough to support the conclusion. Authors can provide a detailed study to support their conclusion. 

Author Response

Please see the attachment - Reviewer 3 comments are listed in blue text under Reviewer 3 heading.

Round 2

Reviewer 3 Report

The authors have done a great job of revising the manuscript and it certainly improved the quality of the article. 

Author Response

Thank you for allowing us to make the changes. Please see the revised manuscript for further alterations.